# BRANCH-GAN: IMPROVING TEXT GENERATION WITH (NOT SO) LARGE LANGUAGE MODELS

**Fredrik Carlsson**[*]  **Johan Broberg**[*]  **Erik Hillbom**[*]  **Magnus Sahlgren**[†]  **Joakim Nivre**[*]
[*]RISE Research Institutes of Sweden     [†]AI Sweden     Correspondence: fredrik.carlsson@ri.se

## ABSTRACT

The current advancements in open domain text generation have been spearheaded by Transformer-based large language models. Leveraging efficient parallelization and vast training datasets, these models achieve unparalleled text generation capabilities. Even so, current models are known to suffer from deficiencies such as repetitive texts, looping issues, and lack of robustness. While adversarial training through generative adversarial networks (GAN) is a proposed solution, earlier research in this direction has predominantly focused on older architectures, or narrow tasks. As a result, this approach is not yet compatible with modern language models for open-ended text generation, leading to diminished interest within the broader research community. We propose a computationally efficient GAN approach for sequential data that utilizes the parallelization capabilities of Transformer models. Our method revolves around generating multiple branching sequences from each training sample, while also incorporating the typical next-step prediction loss on the original data. In this way, we achieve a dense reward and loss signal for both the generator and the discriminator, resulting in a stable training dynamic. We apply our training method to pre-trained language models, using data from their original training set but less than $0.01\%$ of the available data. A comprehensive human evaluation shows that our method significantly improves the quality of texts generated by the model while avoiding the previously reported sparsity problems of GAN approaches. Even our smaller models outperform larger original baseline models with more than 16 times the number of parameters. Finally, we corroborate previous claims that perplexity on held-out data is not a sufficient metric for measuring the quality of generated texts.

## 1   INTRODUCTION

The recent rapid advancements in AI have led to text generation capabilities that, according to some, result in human quality texts (Bubeck et al., 2023). This advancement has primarily been propelled by the foundational role of Transformer-based large language models (LLMs) (Radford et al., 2019; Brown et al., 2020; Chowdhery et al., 2022; OpenAI, 2023; Penedo et al., 2023; Touvron et al., 2023a;b; Biderman et al., 2023). The Transformer architecture (Vaswani et al., 2017) processes sequential data in parallel, in contrast to the previously popular RNNs (Rumelhart et al., 1986; Hochreiter & Schmidhuber, 1997). This computational efficiency has enabled the scaling of both model parameters and training data, leading to enhanced performance and emergent capabilities.

However, various sources report deficiencies in modern LLMs, such as repetitive texts, looping issues, and a lack of robustness (Fu et al., 2021; Holtzman et al., 2020; Wang et al., 2022). One hypothesis attributes these issues to *exposure bias* (Bengio et al., 2015; Ranzato et al., 2016), which highlights the disparity between next token prediction during training, and multi-step generation during inference. Since each generation step risks being erroneous, inference is susceptible to accumulating errors that progressively drift the model away from the training distribution. Although heuristics and sampling strategies reduce the likelihood of erroneous individual generation steps (Keskar et al., 2019; Holtzman et al., 2020), they address symptoms rather than the underlying root cause.

Theoretically, training language models via generative adversarial networks (GANs) (Goodfellow et al., 2020) provides an elegant solution to exposure bias. Training with multi-step generation offers the generator a learning signal more akin to inference, from which it can learn to correct after

any misstep during generation. Unfortunately, GANs come with their own set of challenges (Caccia et al., 2020). Multi-step generation increases the computational burden and the volatile training dynamic can lead to non-convergence and model collapse. Earlier work has addressed training dynamic issues using methods like dense rewards (Shi et al., 2018; Semeniuta et al., 2019; de Masson d'Autume et al., 2019) and incorporating next-token prediction, also known as maximum likelihood estimation (MLE). However, this line of research has, to the best of our knowledge, focused mainly on RNNs, and is hence incompatible with the computational parallelism of Transformers.

We propose the computationally efficient **Branch-GAN**, which utilizes the parallelization capabilities of the Transformer architecture. Our method is based on generating multiple branching sequences from each training sample, which are processed in parallel for both the generator and the discriminator. This parallelization enables massively dense rewards for both the generated and original texts, resulting in an unproblematic and stable training dynamic. According to human evaluation, LLMs utilizing the Branch-GAN method generate significantly better texts on average. Finally, we note a strong correlation between human preferences and automatic metrics such as TTR, with Branch-GAN improving on these metrics without an explicit incentive.

Our main contributions are the following:

- A new method for training Transformers in a GAN setup, which gives higher quality in text generation with fewer model parameters
- A dataset with manual rankings of text quality containing more than 10,000 annotations
- An exploration of automatic metrics that correlate well with human perceived text quality
- Implementation at: Github.com/FreddeFrallan/Branch-GAN

## 2 RELATED WORK

In the realm of GANs for language modeling, most methods were developed prior to the widespread adoption of pre-trained Transformers and therefore use RNNs (Yu et al., 2017; Che et al., 2017; Lin et al., 2017; Guo et al., 2018; Shi et al., 2018; de Masson d'Autume et al., 2019). As a result, they cannot compete with the generative capabilities of modern LLMs. More recently Scialom et al. (2020), Wu et al. (2021), Scialom et al. (2021), Lamprier et al. (2022) trained language GANs or cooperative language GANs with pre-trained Transformers architetures, using models such as GPT-2 (Radford et al., 2019) as generators and RoBERTa (Zhuang et al., 2021) as discriminators, or with encoder-decoder models such as T5 (Raffel et al., 2020) and BART (Lewis et al., 2019). While these advancements have pushed the boundaries of language GANs, many of the challenges historically associated with GANs for language modeling still remain relevant today.

A critical issue in the adaptation of GANs for sequential data is the sparsity of the loss signal. In contrast to traditional next-token prediction models, where loss is computed at each token, the straightforward application of GANs in language modeling typically yields loss only for fully generated sequences. To address this challenge, some methods leverage RNN-based discriminators to process sequences incrementally, thereby providing token-level loss feedback (Shi et al., 2018; Semeniuta et al., 2019; de Masson d'Autume et al., 2019). Alternatively, work such as Lin et al. (2017) modifies the discriminator's classification task to a ranking task, allowing them to rescale rewards from different samples in order to offer a more informative signal for the generator.

Additionally, the inherent instability of GAN training is further exacerbated by the gradient estimation step of reinforcement learning (RL) (Sutton & Barto, 2018). Though some methods train language GANs from scratch (Che et al., 2017; de Masson d'Autume et al., 2019), the majority of methods resort to MLE pretraining as a foundational step (Yu et al., 2017; Shi et al., 2018; Nie et al., 2019; Guo et al., 2018; Scialom et al., 2020; 2021). Approaches such as Che et al. (2017) and Guo et al. (2018) combine MLE and adversarial training to effectively stabilize the learning process.

Other ways to address the instability issues is by averaging rewards across multiple generated samples, as demonstrated by Yu et al. (2017), or by employing a moving average of rewards over time steps (de Masson d'Autume et al., 2019). In a different vein, both Shi et al. (2018) and Wu et al. (2021) recast text generation as an inverse reinforcement learning problem in an effort to increase stability. Their objective is to learn a reward function that rationalizes an "expert policy" responsible for generating the training data. Yet another approach uses the decoder in a cooperative decoding scheme to produce more realistic samples (Scialom et al., 2021; Lamprier et al., 2022).

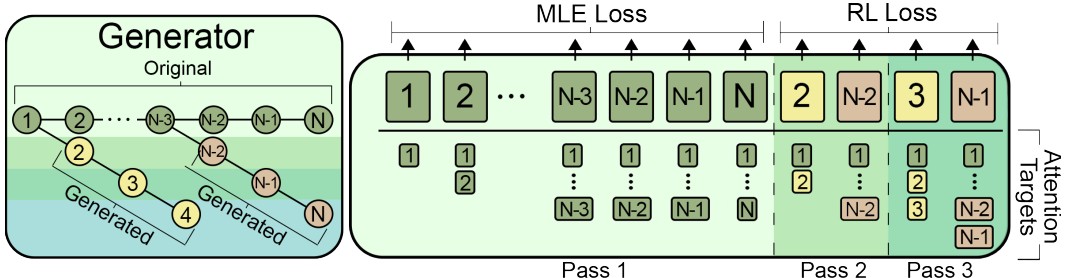

Figure 1: The Branch-GAN generator outputs multiple branching sequences from the original text (left) and is trained using a combination of standard MLE loss on the original text and RL loss on branching sequences (right).

## 3 METHOD

The key idea behind Branch-GAN is to efficiently generate multiple short continuations that branch out from an original text sequence, and supply a dense loss signal for both the original and generated sequences. The generator is trained via both MLE on the original sequence and an adversarial signal from the discriminator for the generated sequences. Simultaneously, the discriminator is trained to better distinguish original sequences from generated sequences.

Our method relies on the auto-regressive Transformer architecture for both the generator and the discriminator. As visualized in Figure 1, we apply a branching self-attention schema to partly process multiple branching sequences in each forward pass. This schema allows both the generator and the discriminator to efficiently compute independently generated sequences, and results in a dense learning signal for both models.

In the following sections, $S$ represents an original text sequence in the training data with length $N$. We let $G$ denote a generated sequence with length $d$ where $d$ is referred to as depth. Finally, we use subscript notation to refer to a token in a sequence, meaning that $S_n$ refers to the $n$:th token in $S$.

### 3.1 GENERATOR

During training we randomly sample $K$ unique starting points in the range $[1, N-1]$ for each batch, determining the start indices for our generated sequences $\{G^1, G^2 ... G^K\}$. From an initial forward pass over $S$ we greedily pick the first token for each generated sequence, but block the actual continuation token in $S$. This means that if the first starting index $K^1 = n$, then $G_0^1 \neq S_{n+1}$.

Following this, we perform $d - 1$ forward passes, greedily selecting and appending a new token to each generated sequence. During these passes, each token self-attends to all previous tokens in the same generated sequence, in addition to all tokens in the original $S$ that lie before its start index. Thus, the n:th token in $G^i$ attends to $\{S_1 ... S_{K^i}\} + \{G_1^i ... G_n^i\}$.

In this way, each forward pass creates $K$ new tokens. After $d$ forward passes we end up with the original sequence $S$ together with $K$ generated branching sequences consisting of $d$ tokens each. The generator is trained via causal language modelling loss over $S$, and via an adversarial RL loss for tokens in the generated sequences (Section 3.3).

### 3.2 DISCRIMINATOR

The discriminator processes data in a fashion similar to the generator (Section 3.1), but with an additional forward pass for the final generation step. This means that an original text $S$ and $K$ branching sequences with depth $d$ requires $d + 1$ forward passes, regardless of the number of sequences $K$. A visualization of the discriminator and its attention is available in Appendix A.

At each time-step $n$, the discriminator provides three scalar predictions $D_n$, $I_n$ and $B_n$, where $D_n$ is the discriminative prediction of the $n$:th token, while $I_n$ and $B_n$ are value head predictions for the $n+1$ time-step. All three of these predictions are passed through a logistic function and thus fall in the range $[0, 1]$.

The discriminative prediction will, due to the sequential nature of the task, be influenced by its pretext. For example, if the discriminator at a point identifies a sequence as 'fake', it will influence subsequent predictions negatively. Thus, the generator RL loss is derived using the discriminator's value heads. The value head prediction $I_n$ estimates $D_{n+1}$ when the next token is from $S$. And $B_n$ estimates $D_{n+1}$ if the next token is from $G$. The RL loss is described in Equation 1.

During training, we can only assign labels to value heads when the correct information is available in the next time step. Thus, $I_n$ is assigned labels at steps found within the original sequence $S_{1,N-1}$, while $B_n$ is assigned labels during the generated sequences $G_{1,d-1}$, and the original time-steps $S_n$ where $n + 1 \in K$, indicating the start of a generated sequence at the subsequent step.

## 3.3 GAN Setup

As is common with GANs, the generator is trained to produce $G$ to fool the discriminator, and the discriminator is trained to better distinguish between $S$ and $G$. The discriminator utilizes binary cross-entropy for both the prediction $D_n$ and the value head predictions, leading to the discriminate loss $\mathcal{L}_D$ and the value head loss $\mathcal{L}_V$. The final discriminator loss is the weighted sum of $\mathcal{L}_D + \mathcal{L}_V * \alpha$.

The generator receives a normal MLE loss on predictions for $S$, along with an RL loss for generated time-steps in $G$ that do not exceed the position $N$. Time-steps in $G$ are classified into one of four categories in accordance with Equation 1, and given a loss $\mathcal{L}_G$ accordingly. The final generator loss is the mean loss over all time-steps, for both $S$ and $G$.

$$
\mathcal{L}_G = \begin{cases} 0 & \text{if } I_{t-1} \leq B_{t-1} \\ -\log(p(t)) & \text{if } I_{t-1} \leq D_t \\ -\frac{I_{t-1}}{D_t - B_{t_1}}\log(p(t)) & \text{if } B_{t-1} \leq D_t < I_{t_1} \\ -\log(1 - p(t)) & \text{if } D_t < B_{t-1} \end{cases} \tag{1}
$$

## 4 Experiments and Analysis

In this section, we evaluate text generation using models trained with Branch-GAN, as well as several baseline models. We begin with a large-scale human evaluation of text quality (Section 4.2), explore automatic metrics that correlate with human quality judgments (Section 4.3) and use these metrics in an automatic model evaluation (Section 4.4). We conclude with a robustness evaluation (Section 4.5) and an ablation study focusing on the hyperparameters depth and sparsity (Section 4.6).

### 4.1 Experimental Setup

Although not a necessity, our training starts from pre-trained language models, where both the generator and the discriminator are initialized from the same checkpoint. We use checkpoints from the Pythia model suite (Biderman et al., 2023), which were trained on the full Pile dataset (Gao et al., 2020) using MLE training in accordance with the current LLM scaling laws (Kaplan et al., 2020).

In order to make our models comparable to the existing Pythia models we train exclusively on data from the Pile dataset. Our training data consists of 100k randomly selected sequences of length 128, tokenized using the Pythia tokenizer. As the full deduplicated version of the Pile contains about 207B tokens and we use 12.8M tokens, less than 0.01% of the Pile data is used during our training.

We train using a branch sequence depth of $d$=16 and $K$=32 branches per sample. This is performed with a batch size of 8, for 4 epochs over the training data, resulting in 50k optimizer updates, where $\frac{4}{5}$ of the tokens come from generated sequences. The discriminator loss weight is set to $\alpha = 0.2$.

Generating text requires the specification of a sampling method and its hyperparameters. Corroborating earlier work, we found baseline models to generate more fluent texts when using top-$p$ sampling (Holtzman et al., 2020) coupled with a repetition penalty. Thus, Pythia and LLama-2 7B use top-$p$=0.95 sampling, unless otherwise stated. Branch-GAN models instead use greedy decoding (unless otherwise stated), since that is what is used during training. Models marked with an asterisk ($*$), use a repetition penalty of 1.2 as proposed by Keskar et al. (2019), and all models utilize a beam size of 8. Results and analyses of additional sampling methods are available in Appendix E.2.

Table 1: Human evaluation results for models in comparison to original texts ($Wiki$, $Fiction$) and to Branch-GAN 1B* (greedy) ($Wiki^{\text{BG-1B*}}$, $Fict^{\text{BG-1B*}}$). Numbers represent the percentage of texts that are judged to be better than or as good as the competing candidate.

| Model | $Wiki$ | $Wiki^{\text{BG-1B*}}$ | $Fiction$ | $Fict^{\text{BG-1B*}}$ |
|---|---|---|---|---|
| *Strong Baselines* | | | | |
| LLama-2 7B (top-$p$) | 32.0 | 50.7 | 38.7 | 43.3 |
| GPT-4 | 81.0 | 88.3 | 73.3 | 86.7 |
| *Comparative Baselines* | | | | |
| Pythia 410M* (top-$p$) | 10.7 | 13.0 | 6.7 | 11.3 |
| Pythia 1B* (top-$p$) | 15.7 | 23.3 | 11.7 | 16.0 |
| Pythia 6.9B* (greedy) | 14.0 | 16.7 | 22.7 | 38.0 |
| Pythia 6.9B* (top-$p$) | 24.0 | 38.7 | 27.3 | 45.7 |
| *Our Models* | | | | |
| Branch-GAN 410M (greedy) | 49.7 | 58.7 | 51.7 | 68.0 |
| Branch-GAN 410M* (greedy) | 48.3 | 57.3 | 71.0 | **70.7** |
| Branch-GAN 1B* (top-$p$) | 48.7 | **65.7** | 50.3 | 59.3 |
| Branch-GAN 1B* (greedy) | **68.7** | - | **76.0** | - |

## 4.2 HUMAN EVALUATION

To thoroughly evaluate text quality, we conduct a rigorous evaluation study, gathering over 10k human annotations from verified English speakers. To ensure high-quality annotations, each annotator was independently verified and hired as a freelancer[1] and fairly compensated. More information regarding the annotation process and the resulting dataset is available in Appendix B.

The evaluation is centered around deciding the preferred continuation to a given context, and posed in two different scenarios. In the first scenario, one continuation is always the original text from the corresponding context. In the second scenario, one continuation always come from Branch-GAN 1B* (greedy). The annotators' task is to determine the preferred continuation, or classify them as equally good.

Each trained model is evaluated on both scenarios and on contexts from two different datasets: Wikipedia (Merity et al., 2017) and Fictional Stories.[2] We randomly select 100 samples from each dataset with a length of 128 tokens, of which 32 are used as context and 96 as continuation. Each model generates continuations to all contexts and we gather 3 annotations per continuation comparison.

Results in Table 1 display the percentage of times that generated texts are either preferred to or judged as equally good as the other candidate. Branch-GAN outperforms all baselines with significant margins, with the exception of GPT-4. Both Branch-GAN and the Pythia models perform better on the fictional contexts compared to the more factual Wikipedia. The opposite is true for GPT-4, which is only just as good as Branch-GAN on fictional stories. Applying repetition penalty to Branch-GAN only improves its score on fictional stories.

---

[1]https://www.upwork.com/
[2]https://www.kaggle.com/datasets/jayashree4/fiction

Table 2: Fine-grained human evaluation for a subset of models when compared against original texts. All Pythia models use top-$p$ sampling; all Branch-GAN models use greedy decoding.

| Model | *Wiki* | | | *Fiction* | | |
|-------|------|------|--------|------|------|--------|
| | **Wins** | **Ties** | **Losses** | **Wins** | **Ties** | **Losses** |
| *Baselines* | | | | | | |
| Pythia 1B* | 9.3 | 6.7 | 84.0 | 4.7 | 7.0 | 88.3 |
| Pythia 6.9B* | 9.3 | 14.7 | 76.0 | 10.7 | 16.7 | 72.7 |
| GPT-4 | 41.3 | 39.7 | 19.0 | 37.3 | 30.7 | 32.0 |
| *Our Models* | | | | | | |
| Branch-GAN 410M* | 14.7 | 36.3 | 49.0 | 18.3 | 52.7 | 29.0 |
| Branch-GAN 1B* | 11.7 | 54.3 | 34.0 | 18.3 | 57.7 | 24.0 |

Table 2 denotes the fine-grained annotations for models when compared to the original texts. Branch-GAN 1B* scores over 50% ties for both datasets, while the Pythia models never tie more than 20%. GPT-4 achieves considerably more wins than other models, but also over 30% losses on the Fiction dataset. These losses may be due to annotators' personal preference in style. More fine-grained results are available in Appendix E.1.

### 4.3 HUMAN PREFERENCES AND AUTOMATIC METRICS

While the human evaluation shows that adversarial training improves the models' ability to generate texts in accordance with human preferences, it does not tell us what properties are characteristic of such texts. As a starting point to finding automatic evaluation metrics that correlate with perceived text quality we investigate two properties: Type-Token Ratio (TTR) and cross-entropy[3] of Pythia 6.9B. The idea is that TTR can capture the repetitiveness of a text, while cross-entropy informs us about the flow of the text.

We calculate these characteristics for all generated texts that received full annotation consensus, and compare them to the characteristics of the original human written texts. As in Section 4.2, an annotation is considered good if it is either preferred to or deemed equally good as the alternative continuation. As detailed in Appendix B, 56% of the samples had agreement from all three annotators.

As seen in Figure 2, generated texts with similar TTR and cross-entropy as that of human written texts are far more likely to receive positive annotations. There exists virtually no texts unanimously rated as good with a TTR $< 0.5$. The span of cross-entropy for good texts is a bit wider, although still clearly centered around the mean for human texts. These results strengthen our intuition that characteristics of generated texts should be similar to those of human written texts, and that such characteristics can be partially captured using a combination of metrics like TTR and cross-entropy.

---

[3]Cross-entropy was preferred over perplexity, as it is easier to visualize.

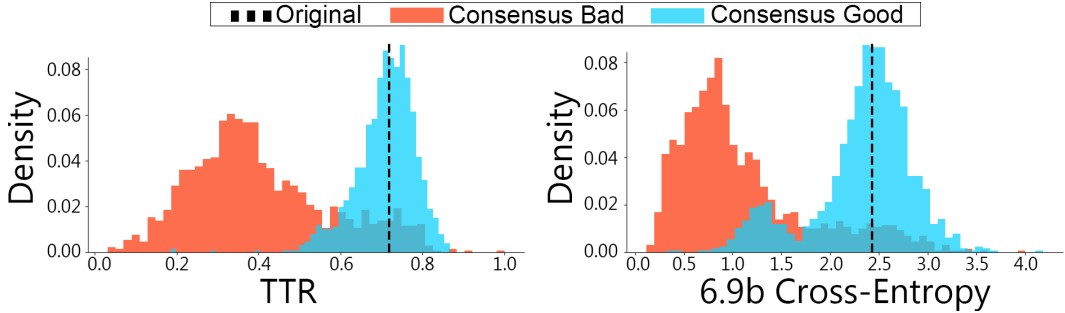

Figure 2: Distributions of TTR and Cross-entropy for Pythia 6.9B, for generated texts with annotation consensus. Dotted lines indicate the mean TTR and Cross-entropy of the original texts.

Table 3: Automatic evaluation of models using perplexity of original text ($Org\ PPL$), perplexity of Pythia 6.9B* on generated texts ($6.9b\ PPL$), type-token ratio ($TTR$), and Self-BLEU ($S - BLEU$).

| Model | $Org\ PPL$ | $6.9b\ PPL$ | $TTR$ | $S - BLEU$ |
|---|---|---|---|---|
| *Ground Truth* | | | | |
| Original Texts | - | 13.08 | 71.8 | 2.4 |
| *Comparative Baselines* | | | | |
| Pythia 1B* | 20.34 | 2.0 | 36.5 | 4.0 |
| Pythia 6.9B* | 13.08 | 1.96 | 45.6 | 3.9 |
| *Our Models* | | | | |
| Branch-GAN 1B | 51.47 | 9.78 | 70.0 | 1.5 |
| Branch-GAN 1B* | 51.47 | 10.38 | 73.8 | 1.4 |
| *Strong Baselines* | | | | |
| LLama 7B | 10.34 | 3.81 | 54.2 | 1.3 |
| GPT-4 | - | 15.4 | 74.9 | 2.0 |

## 4.4 AUTOMATIC EVALUATION

Working from the findings in Section 4.3, we proceed to investigate the characteristics of generated texts from individual models. In addition to measuring TTR and perplexity, we also measure the Self-BLEU score, along with the model's perplexity on the original text continuations. The Self-BLEU score is a measure of diversity between all generated texts, where lower indicates more diversity. Further details and information on these metrics can be found in Appendix D.

In Table 3 we note that Branch-GAN displays a much higher perplexity for the original texts than the baseline models. Texts generated by the Pythia models exhibit significantly lower perplexity and TTR compared to the original texts. In contrast, texts produced by Branch-GAN demonstrate TTR and perplexity scores that are more aligned with the original texts, with the repetition penalty elevating both scores. As indicated by Branch-GAN's lower Self-BLEU score, texts from Branch-GAN are more diverse than other baselines, and even the original texts. Interestingly, GPT-4 has both higher perplexity and lower diversity than Branch-GAN, while matching the TTR value of both Branch-GAN and the original texts. By contrast, the LLama model has high diversity, intermediate TTR and low perplexity, making it more similar to the Pythia models except for diversity.

## 4.5 ROBUSTNESS

To quantify the robustness effects of our adversarial training, we investigate the models' ability to generate texts from varying levels of noisy contexts. We create noise via character repetitions, where randomly selected characters are repeated before tokenization. For these experiments we report how TTR and perplexity according to Pythia 6.9B's change for the generated texts, as noise in the context increases.

As with previous experiments the context length is set to 32 tokens (after noise is introduced), and the generated texts are 96 tokens. This is done for the combined 100 Wikipedia contexts and 100 Fictional Stories contexts that are used in Section 4.4.

The results in Figure 3 demonstrate that the original Pythia models are susceptible to noise, as the mean TTR drops roughly 0.3 units for even small levels of noise. Branch-GAN, although experiencing an early drop in TTR of roughly 0.1 units, remains fairly stable up until 16 character repetitions. Similarly, the perplexity of texts generated by the Branch-GAN models remain stable for up to 16 character repetitions. After this, Branch-GAN generations not using a repetition penalty start degrading.

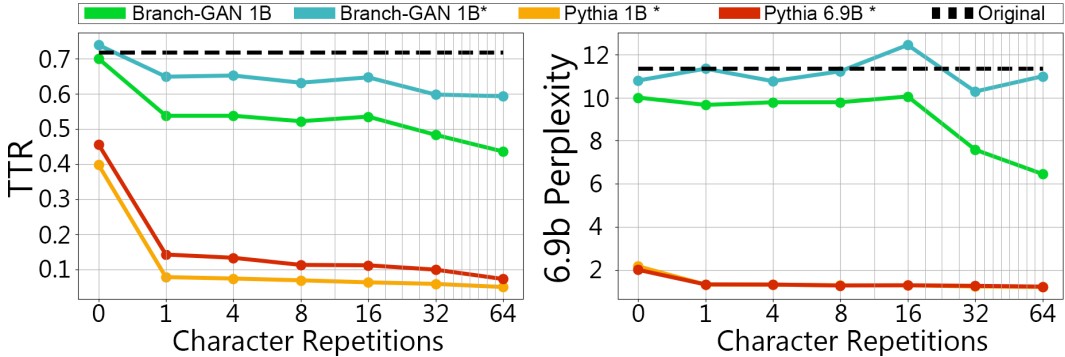

Figure 3: Effects on generated texts when character repetitions are injected into the context.

## 4.6 ABLATION STUDY FOR DEPTH AND SPARSITY

Our final experiment explores the effects of training with different generation depths $d$, and different numbers of branch sequences $K$. For each configuration we train a Branch-GAN 410m for 4 epochs using the setup described in Section 4.1. Using a repetition penalty of 1.2, each model then generates 200 continuations, each 96 tokens long, for the combined Wikipedia and fiction dataset described in Section 4.4.

The first experimental setup keeps the depth $d = 16$ fixed while $K$ changes. The second keeps the $K = 32$ number of branch sequences fixed, while $d$ changes. Finally, since increasing $K$ can be done without increasing the number of forward passes, we keep the number of generated tokens $K * d = 512$ fixed as $d$ changes.

The results in Figure 4 indicate that a certain depth is required to achieve a good TTR, and cannot be compensated for by increasing the number of branches. However, we find diminishing returns for TTR when increasing the depth beyond 8, especially when using repetition penalty. Interestingly, decreasing the number of branches to $(K = 16, d = 16)$, or even $(K = 8, d = 16)$, performs nearly as well as $(K = 32, d = 16)$. Thus, using fewer branches could be a source for future computational efficiency improvements.

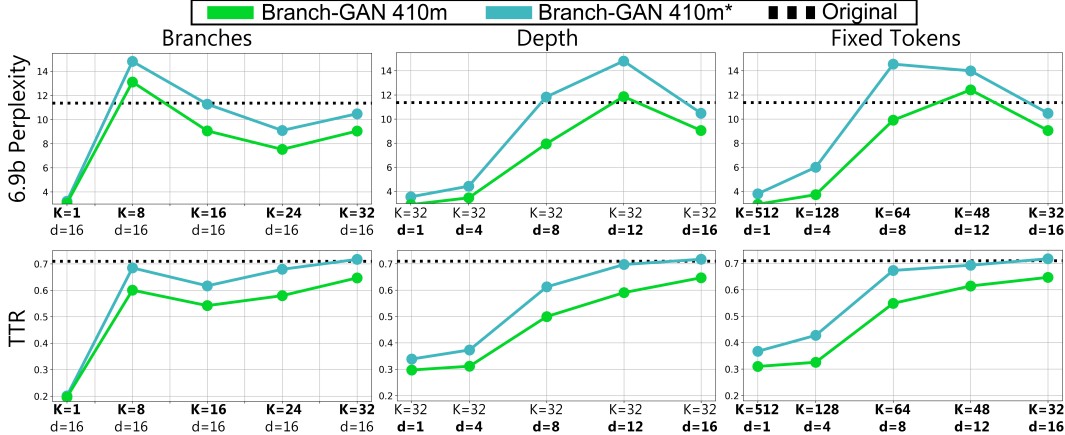

Figure 4: Results from training Branch-GAN 410 for 4 epochs, varying the number of branch sequences $K$ and the depth level $d$. The first row shows Pythia 6.9b's perplexity on the generated texts. The second row shows the TTR of the generated texts.

## 5 DISCUSSION & CONCLUSION

This paper has introduced Branch-GAN, an adversarial learning setup utilizing the parallelization capabilities of Transformers. Although our method is applicable to sequential data in general, we focus on text generation via LLMs. To this end we have conducted a human evaluation study to properly evaluate the textual quality of our method and existing baselines.

Our results clearly indicate that the text generation capabilities of LLMs can be significantly enhanced by incorporating adversarial training. Our Branch-GAN models outperform LLMs that are more than 10 times larger, without training on any new data, alluding to possibly greater gains when applied to larger models.

Interestingly, the Branch-GAN models display comparatively poor perplexity on held-out texts, while generating texts that match TTR and perplexity of texts written by humans. It thus appears that the adversarial training method favors properties that correspond to human preferences at the expense of minimizing perplexity on held-out datasets, something we further analyze in Appendix E.5. However, we argue that any such conclusions require further evidence as our training experiments are performed on very small datasets. Additionally, we note that although the model is only trained on branching sequences of length 16, the beneficial effect generalizes to generation of longer sequences (up to length 96 in our experiments).

The Branch-GAN method opens up for many interesting experiments, such as exploring different sampling methods during training, modifying the relative size of the generator and discriminator, dynamically changing the generation depth, and fine-tuning towards specific downstream tasks. Additionally, we hypothesize that incorporating adversarial learning could drastically increase the data efficiency of LLMs, and encourage further exploration in this direction.

Another line of future research concerns the characterization of text quality using automatic metrics, which we have only started to explore in this paper using TTR, Self-BLEU and cross-entropy. In addition to their use in automatic evaluation, such metrics could potentially also be incorporated into training objectives.

Finally, it is worth noting that many of our training configurations were constrained by our limited computational resources. Therefore, we also encourage further research to investigate the effects of scaling up our proposed method. This includes using larger foundation models and more training data, increasing the generational depth, and training models without relying on additional pre-training.

## ETHICS STATEMENT

The research reported in this paper involves an extensive human evaluation. In the interest of fairness as well as data quality, annotators were hired as freelancers through Upwork[4] and paid 7.5–10 USD for each batch of 100 examples, which corresponds to an hourly rate of 5–10 USD. All datasets used are publicly available, as specified in Sections 4.1 and 4.2. For more information about the Pile, we refer to the published data sheet (Biderman et al., 2022).

## REPRODUCIBILITY STATEMENT

The code used for training and evaluation, along with the human annotations underlying the evaluation in Section 4.2, are available at Github.com/FreddeFrallan/Branch-GAN.

All pretrained LLMs used in this research are publicly available, with the exception of GPT-4, which we have nevertheless included in the interest of comparing to the commercial state of the art, even though these results may not be reproducible at a later date. All datasets used for further training and evaluation are likewise publicly available. Finally, the hyperparameter settings are described in Appendix C to facilitate reproduction of our results.

---

[4]https://www.upwork.com

ACKNOWLEDGMENTS

The research presented in this paper was supported by the Swedish Research Council (grant no. 2022-02909) and by a donation from Meta.

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

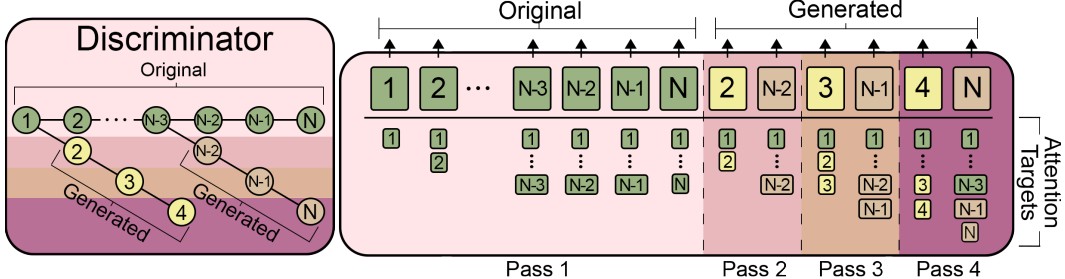

Figure 5: The Branch-GAN discriminator.

# A  DISCRIMINATOR

As described in Section 3.2, The discriminator processes data in a manner similar to that of the Generator. The difference is that it requires an additional step for the final generation pass, as is visualized in Figure 5. The Discriminator makes three predictions at each timestep, as depicted in Figure 6 (left). These predictions are used to construct a loss signal, incorporating the discriminator's current hypotheses about subsequent predictions, as illustrated in Figure 6 (right).

## A.1  DUAL VALUE HEAD LOSS

The intuition behind the Branch-GAN RL loss is derived from the three distinct sections formed by $I_n$ and $B_n$. See figure 6 below. In **Section 1** of the loss, $D_n$ receives a higher discriminate score than the expected score from human-written text $I_{n-1}$. Having successfully fooled the discriminator, the generator is encouraged to simply maximize the log probability of that token. In **Section 2** of the loss, $D_n$ surpasses the expected score for generated text $B_{n-1}$ but falls short of $I_{n-1}$. In this scenario, the log loss is linearly scaled based on its position between $I_{n-1}$ and $B_{n-1}$. In **Section 3** of the loss, $D_n$ receives a score that is lower than the expected score for generated text. As the discriminator is confident that the token is fake, the generator is directed to minimize that token probability. In the corner case that the value heads misalign, and $I_{n-1} \leq B_{n-1}$, the loss is simply discarded. Empirical findings indicate that such misalignments occur extremely rarely during training.

## A.2  OTHER LOSS FUNCTIONS

Early stages of development experimented with more typical loss functions such as REINFORCE Williams (1992). This led to an unstable GAN training dynamic, that often caused a generator collapse. Switching to the dual value head loss mitigated this issue, and since switching zero collapses has been encountered. However, as this was in early stages of the research further work is needed to properly evaluate different loss functions in this GAN scenario.

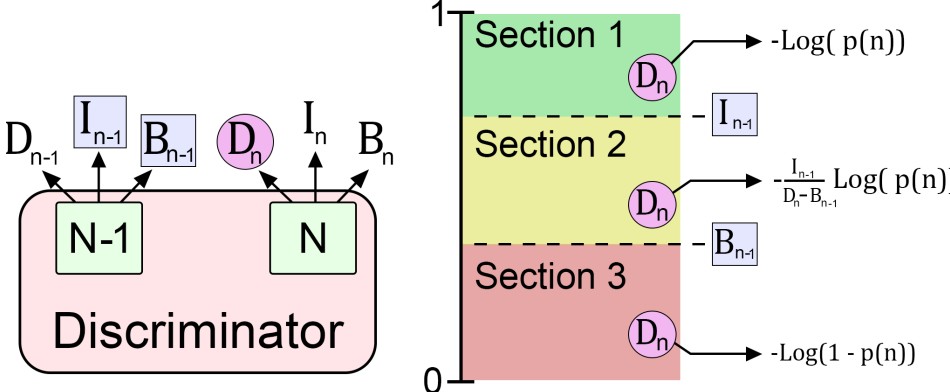

Figure 6: Visualisation of the Discriminator predictions and the RL loss.

# B ANNOTATORS AND COLLECTED DATASET

| Wikipedia and FictionStories | | | |
|---|---|---|---|
| **Description** | **vs. Original** | **vs. Branch-GAN 1B\*** | **Both** |
| Dataset samples | 2000 | 1800 | 3800 |
| Annotators | 9 | 9 | 9 |
| Models | 10 | 10 | 10 |
| Samples per model | 100 | 100 | 100 |
| Annotators per model | 3 | 3 | 3 |
| Evaluated model preferred (%) | 18.3 | 24.1 | 21.1 |
| Evaluated equally good (%) | 23.1 | 23.3 | 23.2 |
| Original/Branch-GAN 1B * preferred (%) | 58.6 | 52.6 | 55.7 |
| Complete agreement[a] (%) | 39.65 | 38.2 | 38.9 |
| Partial agreement[b] (%) | 48.0 | 48.9 | 48.4 |
| Complete disagreement[c] (%) | 12.35 | 12.9 | 12.6 |
| Agreement reliability (Fleiss' Kappa) | 0.22 | 0.26 | 0.24 |

[a] Complete agreement: All three annotators provided the same label.
[b] Partial agreement: Two annotators provided the same label, and one differed.
[c] Complete disagreement: Each annotator provided a different label.

Table 4: Statistics on all the collected annotations. The first column shows statistics for comparisons between models and original texts and the second for comparisons between our 1B parameter Branch-GAN model using repetition penalty of 1.2. The last column shows the statistics for all comparisons.

All hired annotators were provided with the following instruction:

*"Using only your personal preference and no tools such as Google or ChatGPT. Given an initial text snippet and two potential continuations, decide which of the continuations you prefer. Each text and continuation spans at most a couple of sentences.*

*Sometimes the decision is straight forward, as one alternative might be very bad/repetitive. Other times you might be unable to pick one. You can then claim they are of equal quality."*

## C  TRAINING DETAILS

| Hyperparameter | Value |
|---|---|
| *General* | |
| Max Sequence Length | 128 |
| Epochs | 4 |
| *Generator* | |
| Optimizer | Adam |
| Learning rate (start) | $10^{-8}$ |
| Learning rate warmup steps | 1000 |
| Learning rate (end) | $10^{-5}$ |
| Branches $K$ | 32 |
| Depth $d$ | 16 |
| Sampling strategy | Top-$k$ |
| $k$ (in Top-$k$) | 50 |
| Section weight 1 (RL Advantage) | 1 |
| Section weight 2 (RL Advantage) | 1 |
| Section weight 3 (RL Advantage) | 1.5 |
| *Discriminator* | |
| Optimizer | Adam |
| Learning rate | $10^{-4}$ |
| Value head loss weight $\alpha$ | 0.2 |

Table 5: Hyperparameters for Branch-GAN

### C.1  COMPUTATIONAL COST

The computational burden of the Branch-GAN training is highly dependent on the training hyper-parameters. Each increase in depth infers an additional forward pass per sample, whilst the number of branching sequences affects directly affects the memory load. On top of this both the generator and discriminator need to be trained.

The big upside to Branch-GAN is its data efficiency, seeing that the our results were attained with less than 0.01% of the pre-training data. Thus, although our experiments were executed with a straight-forward Python PyTorch, Branch-GAN 1B comfortably completed 4 epochs under 40 hours. The hardware for this training was a single DGX machine with 8 NVIDIA A100-SXM4-40GB GPUs.

# D AUTOMATIC EVALUATION METRICS

## D.1 PERPLEXITY

When evaluating perplexity of models we always provide 32-tokens long contexts from 100 randomly selected samples from both the Wikipedia and Fictional Stories datasets. For each context the model subsequently generates a sequence of 96 tokens. We compute perplexity exclusively on these 96 generated tokens and average the results for all samples.

## D.2 TTR

The type-token ratio (TTR) is the number of unique (sub)words divided by the length of the generated sequence. TTR is a commonly used measure of lexical richness in the language acquisition literature. The unnormalized TTR has been criticized for not being comparable across texts of different lengths. This is not a problem in our case, since all sequences are of the same length.

## D.3 SELF-BLEU

We compute the Self-BLEU metric as described by Zhu et al. (2018), using 32-token long contexts from both the Wikipedia and Fictional Stories datasets. For our evaluation, we randomly select 100 samples from each dataset, leading to a total of 200 samples. Each sample is then used to generate a sequence of 96 tokens. The reported Self-BLEU scores represent the average computed over these 200 generated sequences.

## D.4 CHARACTER REPETITION

When distorting input texts we do so by repeating randomly selected characters in the contexts before tokenization. Any character in the context string may be selected but we prevent characters from being repeated more than ones. Since repeated characters affect the tokenized sequence length (usually making it longer) we truncate the distorted sequence to 32 tokens, the same lengths as the original context. Below is an example of an original context and three different levels of distortions with one, two and four repetitions.

**Original Wikipedia Context:**
Jive Records without the band's authorization or consent. Back 2 Base X peaked at No. 12 on the Independent Albums chart, and at No. 154 on the Billboard 200. Allmusic's Rob Theakston wrote

**Number of repetitions: 1**
Jive Records without the band's authorization or consent**t** . Back 2 Base X peaked at No. 12 on the Independent Albums chart, and at No. 154 on the Billboard 200. Allmusic's Rob Theakston wrote

**Number of repetitions: 2**
Jive Records without**t** the band's authorization or consent**t**. Back 2 Base X peaked at No. 12 on the Independent Albums chart, and at No. 154 on the Billboard 200. Allmusic's Rob Theakston wrote

**Number of repetitions: 4**
Jive Records without**t** the band's authorization or consent**t**. Back 2 Bas**s**e X peaked at No. 12 on the Independent Albums chart, and at No.**.** 154 on the Billboard 200. Allmusic's Rob Theakston wrote

# E ADDITIONAL EXPERIMENTS & RESULTS

## E.1 FINE-GRAINED HUMAN EVALUATION

Table 6 shows the fine-grained annotations for models compared to original texts. Table 7 shows the fine-grained annotations for models when compared against texts generated by Branch-GAN 1B*.

Table 6: Fine-grained human evaluation when compared against original texts. All Pythia models and LLama-2 use top-$p$ sampling; all Branch-GAN models use greedy decoding.

| Model | $Wiki$ | | | $Fiction$ | | |
|---|---|---|---|---|---|---|
| | Wins | Ties | Losses | Wins | Ties | Losses |
| *Baselines* | | | | | | |
| Pythia 410* | 6.7 | 4.7 | 88.7 | 3.3 | 3.3 | 93.3 |
| Pythia 1B* | 9.3 | 6.7 | 84.0 | 4.7 | 7.0 | 88.3 |
| Pythia 6.9B* | 9.3 | 14.7 | 76.0 | 10.7 | 16.7 | 72.7 |
| LLama-2 7B* | 22.3 | 9.7 | 68.0 | 34.3 | 4.3 | 61.3 |
| GPT-4 | 41.3 | 39.7 | 19.0 | 37.3 | 30.7 | 32.0 |
| *Our Models* | | | | | | |
| Branch-GAN 410M* | 14.7 | 36.3 | 49.0 | 18.3 | 52.7 | 29.0 |
| Branch-GAN 1B* | 11.7 | 54.3 | 34.0 | 18.3 | 57.7 | 24.0 |

Table 7: Fine-grained human evaluation when compared against texts generated by Branch-GAN 1B*. Pythia models and LLama-2 use top-$p$ sampling; Branch-GAN model use greedy decoding.

| Model | $Wiki$ | | | $Fiction$ | | |
|---|---|---|---|---|---|---|
| | Wins | Ties | Losses | Wins | Ties | Losses |
| *Baselines* | | | | | | |
| Pythia 410* | 5.0 | 8.0 | 87.0 | 6.7 | 4.7 | 88.7 |
| Pythia 1B* | 10.3 | 13.0 | 76.7 | 12.0 | 6.3 | 81.7 |
| Pythia 6.9B* | 19.0 | 19.7 | 61.3 | 10.7 | 35.0 | 54.3 |
| LLama-2 7B* | 40.3 | 10.3 | 49.3 | 30.3 | 13.0 | 56.7 |
| GPT-4 | 55.3 | 33.0 | 11.7 | 52.3 | 34.3 | 13.3 |
| *Our Models* | | | | | | |
| Branch-GAN 410M* | 28.0 | 29.3 | 42.7 | 17.3 | 53.3 | 29.3 |

## E.2 ADDITIONAL SAMPLING METHODS

We limited our hyperparameter testing for the baseline models with human evaluation, starting with configurations recommended in previous studies. For Pythia 6.9b, we assessed three setups: repetition penalty 1.2 with greedy decoding Keskar et al. (2019); repetition penalty 1.2 with top-$p$=0.9 sampling (Holtzman et al., 2020); and no repetition penalty with greedy decoding. Table 8 shows that repetition penalty and top-$p$ sampling yields the most favorable results.

Table 8: Human evaluation results for different sampling settings for Pythia 6.9B in comparison to original texts ($Wiki$, $Fiction$) and to Branch-GAN 1B* ($Wiki^{\text{BG-1B*}}$, $Fict^{\text{BG-1B*}}$). Numbers represent the percentage of texts judged to be better than or as good as the alternative candidate.

| Model | $Wiki$ | $Wiki^{\text{BG-1B*}}$ | $Fiction$ | $Fict^{\text{BG-1B*}}$ |
|---|---|---|---|---|
| Pythia 6.98B (greedy) | 8 | 24.4 | 20.4 | 25.3 |
| Pythia 6.9B* (greedy) | 14.0 | 16.7 | 28.3 | 38.0 |
| Pythia 6.9B* (top-$p$) | **24.0** | **38.7** | **27.4** | **45.7** |

### E.3 Additional Datasets

To complement the human evaluation performed on Wikipedia and Fictional Stories, we supply automatic evaluation on 5 additional datasets. These datasets were selected to cover a wide range of different domains and consists of: CC-NewsHamborg et al. (2017), Dialogsum Chen et al. (2021), Legal Summaries [5], Medical Transcriptions[6], and OpenWeb Gokaslan & Cohen (2019).

Similar to all previous tasks, each dataset is chunked into into segments of 128 tokens, from which we sample 100. These are then split into 32 tokens pretext and 96 tokens continuations. Each model generates 96 continuation tokens, from which calculate the perplexity and TTR on these. This allows us to compare the metrics of the generated texts against that of the original continuations.

Table 9 shows the perplexity on these texts according to Pythia 6.9B. For each model we show Pythia 6.9B's perplexity on the original texts minus the perplexity on the generated texts. For all datasets, the Branch-GAN models clearly generate texts with perplexity closer to that of the original texts. The biggest difference are for the News and Wikipedia texts, and the smallest difference is for the dialog and fiction texts. For both Branch-GAN models top-$p$ sampling yields a lower difference.

Table 10 shows the TTR, and for each model we show the original TTR minus the TTR of the generated texts. Again, the Branch-GAN models generate texts that have TTR significantly closer to that of the original texts. Interestingly, the biggest difference in TTR are for Dialogsum and Fiction texts, and the smallest difference is for news texts. This is almost a completely reversed ordering compared to the perplexity results. However, just as with the perplexity results, Branch-GAN models using top-$p$ sampling resulted in slightly better results.

Table 9: Perplexity Scores According to Pythia 6.9B

| Model | Legal | OpenWeb | Dialogsum | Medical | News | Wikitext | Fiction |
|---|---|---|---|---|---|---|---|
| *Original Perplexity* | | | | | | | |
| Original Texts | 8.67 | 9.41 | 7.25 | 11.52 | 14.03 | 14.79 | 11.37 |
| *Difference In Perplexity Compared To Original* | | | | | | | |
| Pythia 410m (g) | 7.12 | 7.56 | 6.02 | 9.95 | 12.20 | 12.89 | 9.66 |
| Pythia 410m* (g) | 7.03 | 7.26 | 5.98 | 9.69 | 11.81 | 12.66 | 9.44 |
| Pythia 410m* (s) | 6.85 | 7.09 | 5.94 | 9.59 | 11.53 | 12.48 | 9.20 |
| Pythia 1B (g) | 7.09 | 7.74 | 6.05 | 10.00 | 12.12 | 12.90 | 9.63 |
| Pythia 1B* (g) | 7.00 | 7.38 | 6.01 | 9.81 | 11.80 | 12.65 | 9.43 |
| Pythia 1B* (s) | 6.98 | 7.21 | 5.94 | 9.62 | 11.65 | 12.57 | 9.22 |
| Br-GAN 410m (g) | 3.36 | 1.41 | 4.11 | 4.82 | 5.56 | 5.68 | 1.61 |
| Br-GAN 410m* (g) | 1.84 | -0.42 | 4.03 | 3.31 | 4.55 | 4.56 | 0.45 |
| Br-GAN 410m* (s) | 1.16 | **0.39** | 2.95 | **2.26** | **3.95** | **3.55** | **0.03** |
| Br-GAN 1B (g) | 1.82 | 1.63 | 2.98 | 4.86 | 5.36 | 5.24 | 1.37 |
| Br-GAN 1B* (g) | 1.34 | 1.20 | 1.20 | 4.93 | 5.59 | 4.82 | 0.58 |
| Br-GAN 1B* (s) | **0.53** | 1.70 | **-0.05** | 4.01 | 4.85 | 4.34 | 0.07 |

---

[5]https://huggingface.co/datasets/lighteval/legal_summarization
[6]https://huggingface.co/datasets/rungalileo/medical_transcription_4

Table 10: Type Token Ratio

| Model | Legal | OpenWeb | Dialogsum | Medical | News | Wikitext | Fiction |
|---|---|---|---|---|---|---|---|
| *Original TTR* | | | | | | | |
| Original Texts | 66.2 | 74.2 | 61.8 | 68.8 | 73.4 | 71.8 | 71.8 |
| *Difference In TTR Compared To Original* | | | | | | | |
| Pythia 410m (g) | 37.9 | 43.0 | 39.9 | 44.8 | 43.2 | 42.8 | 49.9 |
| Pythia 410m* (g) | 34.8 | 34.0 | 39.0 | 38.3 | 34.4 | 38.3 | 44.3 |
| Pythia 410m* (s) | 28.9 | 33.5 | 37.8 | 36.9 | 30.3 | 36.0 | 40.6 |
| Pythia 1B (g) | 33.2 | 42.2 | 38.9 | 42.3 | 39.2 | 37.8 | 46.1 |
| Pythia 1B* (g) | 29.5 | 26.3 | 38.7 | 35.7 | 30.5 | 30.4 | 40.2 |
| Pythia 1B* (s) | 28.5 | 30.3 | 37.8 | 30.9 | 28.1 | 30.9 | 37.0 |
| Pythia 6.9B (g) | 28.9 | 30.7 | 36.4 | 40.3 | 33.2 | 33.4 | 38.15 |
| Pythia 6.9B* (g) | 24.1 | 20.0 | 37.3 | 33.2 | 25.1 | 25.0 | 32.0 |
| Pythia 6.9B* (s) | 24.2 | 19.4 | 34.6 | 31.7 | 19.6 | 24.6 | 30.3 |
| Br-GAN 410m (g) | 15.2 | 10.6 | 26.4 | 12.6 | 11.5 | 9.8 | 8.0 |
| Br-GAN 410m* (g) | 5.7 | -4.2 | 21.4 | 2.1 | 5.4 | 1.5 | 2.1 |
| Br-GAN 410m* (s) | 4.9 | 3.2 | 16.4 | 1.9 | 2.4 | **0.6** | **1.0** |
| Br-GAN 1B (g) | **0.6** | 3.4 | 14.4 | 5.7 | 4.1 | 1.9 | 1.8 |
| Br-GAN 1B* (g) | -3.6 | -1.4 | **2.5** | **0.7** | -0.1 | -1.9 | -2.1 |
| Br-GAN 1B* (s) | -4.1 | **-0.6** | **-2.5** | -2.3 | **-0.0** | -3.3 | -1.7 |

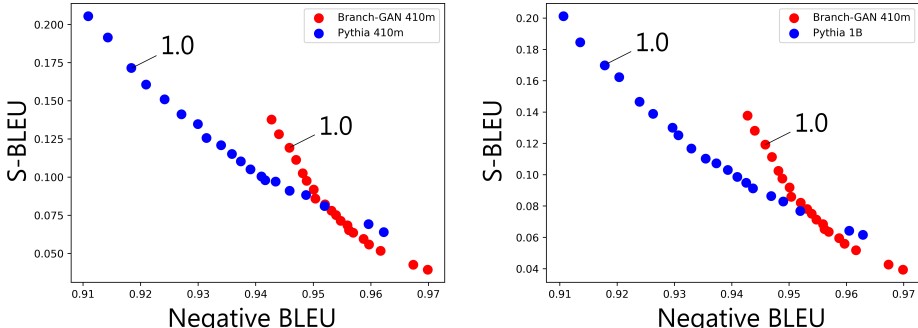

Figure 7: Quality and Diversity scores for different temperatures. Temperature 1.0 as was used during all other experiments is marked for each model

### E.4 TEMPERATURE SWEEP

Caccia et al. (2020) found that although GAN training can perceivably increase textual quality, this may be for a specific temperature, and at the cost of diversity. Therefore they propose an experiment that evaluates quality and diversity over a sweep of different temperatures using automatic metrics.

For each temperature the model generates 10k texts to the contexts of a corpus. Quality is measured via the BLEU overlap (Papineni et al., 2002) of the generated texts and texts from the test set. Diversity is measured using self-BLEU (Section D). The proposed experiment is analyzed using a two-dimensional scatter plot over the two metrics, where the quality metric is inverted. Thus the plot origin is the ideal for both metrics.

We use the EMNLP News-20217 corpus from Subramanian et al. (2017). Similar to the original work we also explored temperatures in range of $[0.9, 4.0]$. Finally, we note that all evaluation was performed using the original published code.[7]

Figure 7 contains the results of this experiment, with temperature 1.0 marked as this is the temperature that was used during all other experiments. The results clearly indicate the both the Pythia models generate texts that overlap more with the test sets. The Pythia models are also able to find temperatures that yield similar quality to that of Branch-GAN's best quality, but with higher diversity.

Interestingly, the quality measurement of this experiment radically contradicts the results attained from our human evaluation study. As clearly demonstrated in Section 4.2, Branch-GAN with a temperature of 1.0 generates texts with a higher quality, according to humans. However, the diversity metrics for temperature 1.0 of both models are aligned with our results in Section 4.4

### E.4.1 AUTOMATIC METRICS FOR OPEN-ENDED TEXT GENERATION

We fully support the importance of evaluation over different sampling temperatures, but we argue against the use of BLEU as an automatic evaluation metric for open-ended text generation. The discrepancy between our human study and the BLEU scores corroborates this stance.

Considering the vast amount of possible texts that are of "high quality", the quality of open-ended text generation is not easily captured by overlap metrics. These metrics have been shown to correlate well with human judgment on constrained tasks such as translation or image captioning. However, to the best of our knowledge, no study has demonstrated their correlation on open-ended text generation.

Admittedly, this is a problem for the field of open-ended text generation as human evaluation is both timely and expensive. Until further work has progressed on automatic metrics we are thus unable to properly evaluate large ranges of temperatures. Conclusively, we cannot disprove that there exists a sampling strategy where the original Pythia models outperform our Branch-GAN models.

---

[7]https://github.com/pclucas14/GansFallingShort

## E.5 NEGATIVE LOG-LIKELIHOOD

Table 11: Average NLL Scores for Pythia and Branch-GAN Models

| | Generated | | Original | | |
| Model | Good Texts ↓ | Bad Texts ↑ | Wiki ↓ | Fiction ↓ | The Pile ↓ |
|---|---|---|---|---|---|
| Pythia 410m | 3.10 | 2.20 | 3.42 | 3.35 | 2.56 |
| Pythia 1B | 2.86 | 2.06 | 3.17 | 3.20 | 2.38 |
| Pythia 6.9B | **2.75** | 1.87 | **2.80** | **2.86** | **2.10** |
| Branch-GAN 410m | 3.57 | 3.10 | 4.14 | 4.03 | 3.15 |
| Branch-GAN 1B | 3.27 | **3.20** | 4.13 | 4.10 | 3.10 |

To accompany the automatic evaluation in Section 4.4 we provide the models negative log-likelihood scores for different datasets. For each dataset we select 1000 texts, and the sequence length of each text is 128 tokens. The datasets fall into two categories: Original and Generated.

The original texts are simply texts taken from the test set of the corresponding dataset. The generated texts, from any possible model, that received either all positive annotations, or all negative annotations. These are the same as used in Section 4.3. Thus the generated texts are split into Good and Bad.

From the results in Table 11 it is clear that the original Pythia models achieve a lower NLL score across all datasets. However, this also includes the bad texts, that demonstrated the lowest NLL score. This is likely due to all the repetitions within the bad texts, making them more predictable. In contrast, the Branch-GAN has a higher overall NLL score, but the bad texts are not much likelier than the good texts.

### E.5.1 INTERPRETATION & DISCUSSION

The original Pythia models are trained to predict the next token given a context. However, even though repetitive text may be easily predictable, such predictability is not necessarily the desired generative behaviour. We argue that the ultimate goal of a good generator is not to predict the most likely next token given a context. Instead, it should produce continuations that appeal to the reader. In cases where the context is repetitive and of low quality, a continuation that can break this predictable pattern and enhance the overall quality is preferred.

The Branch-GAN models have attained their loss signal differently, as they try to produce sequences indistinguishable from real data, regardless of whether the previous generated token was good or bad. This opens up for the possibility of course correction during the training where, after a repetition is encountered, it might still be beneficial to generate something different. This would explain why the Branch-GAN models exhibits a higher NLL score for the bad texts, although they are easily predictable.

Whether these loss functions differ only in intuition, or actually lead to theoretically and practically distinct generators, is beyond the scope of this paper. However, as seen in Section 4.2, Branch-GAN can generate texts preferred by human evaluators, despite performing worse on terms of PPL and NLL on original data. We acknowledge that these are preliminary results and strongly encourage further investigations into this direction of research.

## F    EXAMPLE OUTPUTS

### F.1    BRANCH-GAN 410M    GREEDY    REPPENALTY: 1.2    BEAMSIZE: 8

**Wikipedia Context:** " kiss you " is a song recorded by english-irish boy band one direction for their second studio album, take me home ( 2012 ). it was
**Generation:** written and sung by. the lyrics were inspired from a poem written by geoffrey o'reily in. it tells a story of love gone wrong between a father and a son and how the son forgivingly accepts his father as his real father, while simultaneously punishing him for not accepting his true father as his real father, in order to maintain the illusion of father and son as one and the same, as evidenced in the song lyrics and imagery. contents

.

**Wikipedia Context:** out a t-34 at tugok, also called morisil. e company never reached its blocking position. a strong north korean force surprised and delivered
**Generation:** a surprise counterattack, which resulted in heavy fighting along the entire front line. heavy fighting continued along the entire front line, forcing enemy units to reorganizing and leading to heavy casualties among the troops on both flanks, as well as among the artillery units along with tanks and armored personnel carriers. in addition to heavy fighting and shelling by ground forces and air units, there were reports that at one time or another, at least two enemy tank battal

.

**Wikipedia Context:** with the increased number of civilian casualties compared with the world war i, winston churchill agreed to ware's proposal that the commission also maintain a record of commonwealth civilian
**Generation:** casualties, to enable the committee to take steps to prevent similar tragedies from occurring again in future. following the publication of this report, a number of members of parliament (mps) raised concerns regarding the committee's handling. amongst others, david dimble mp, who chaired a committee meeting at which it considered evidence relating to the number, location and circumstances of civilian casualties during the first world war said: it is deeply regrettable that, following the publication

.

**Fiction Context:** to assume that it was in this artificial underworld that such work as was necessary to the comfort of the daylight race was done? the notion was
**Generation:** repugnant to my sense of humanity; and besides repugnant to my convictions as to the moral principles by which it ought to be conducted, there seemed to me a fundamentally untenable inference to be drawn therefrom. for it seems to me, first of all—and this arises from my reading of shakespeare—of supposing that there was nothing in the underworld but what belonged to the humane order of

.

**Fiction Context:** playing with the other girls and lads, she had fled into the wood, though never so far as this. it was pleasant to be hidden and
**Generation:** yet to be seen at the same time as all these young fellows; for it gave one the opportunity of watching what was going on around them, while at the same moment hearing what they said and doing what they did, without being overheard. this made one feel more kindly towards them all, for it gave them something to whisper about, while at the same moment hearing what they said and doing what they did without being overheard.

.

**Fiction Context:** they came, and then, pointing to the bronze pedestal, i tried to intimate my wish to open it. but at my first gesture towards
**Generation:** it i saw only the shadowy outline and dim light of the figure beneath, for which reason i felt a pang so keen that i began to doubt whether it had really moved at all, or whether it only spun round my brain like a spinning wheel, and whirling round and round incessant circles, like some gigantic spinning wheel which awaits its completion. but as i gazed at that figure beneath my hand it seemed to me

### F.2 BRANCH-GAN 1B GREEDY REPPENALTY: 1.2 BEAMSIZE: 8

**Wikipedia Context:** in mid-november, the jma briefly tracked a weak tropical depression near wake island. the agency also briefly tracked a tropical depression off the coast of vietnam on
**Generation:** "new year's eve and again early on new year's day. over the next several days the jma tracked several other tropical storms as well as several mid-level hurricanes. most of the storms tracked well away from land and did no direct threat to u.s. cities. however, two tropical storms, one mid-level hurricane, one tropical storm near the lesser antilles, two hurricanes near the us west coast, a storm in mexico,

.

**Wikipedia Context:** experts disagree about the number of species in the genus attalea ( broadly defined ). in 1965, dutch taxonomist jan gerard wessels
**Generation:** made the surprising claim that there were only two attalea species. he cited evidence from a variety of sources to substantiate his claim. these sources included fieldwork in the swartz mountains in the u.s. ( where wessels also made field trips as part of his fieldwork ), letters written by attalea to her sister elinor ( elinor wasn't a writer, so elinor wrote these letters ), fieldwork by

.

**Wikipedia Context:** upon drafting a new charter for manila in june 1901, the americans made official what had long been tacit : that the city of manila was not intramuro
**Generation:** and would henceforth be treated as if it were another philippine province, like cebu or zamboanga. in 1902, president mckinley issued executive order no. 10, formal recognition of manila's extraterritorial jurisdiction. the following year, president woodrow wilson formally recognized manila by proclamation. in 1908, congress ratified the treaty whereby the philippines ceded all of their mineral and agricultural rights to the united states in exchange for $15,000,000.

.

**Fiction Context:** yawned and went on along the cliff, and they followed. then he stopped and went back. it was a splendid night, beset
**Generation:** only by clouds that moved restlessly across the sky, as if the very bed itself was restless, shifting from one side of the room to the other, as if the mattress beneath him quaked from time to time. but he lay still, eyes closed, letting his head fall back so that the top of it rested against the stone wall of the cave, just above the level of his

.

**Fiction Context:** the conditions of underground life, and as happy in their way, as the overworld people were to theirs. as it seemed to me, the
**Generation:** wisest course would be for them all to submit to the will, or at least to abide by it–to believe it, and act upon it, as a universal law. for, surely, if they did so, there would be no room left for error; for there would be but one standard of truth–the will itself! and this being so, what could be wiser than obeying it, thinking it right, and doing it

.

**Fiction Context:** day." the she-bear did not answer, but there came a measured crunching sound out of the darkness. she had turned up a bone
**Generation:** and was tearing it apart, smearing the white meat all about with her demented gleam. "ha!" she said. "now what was that noise?" and with that, a huge shadow overtook her, blotting out the moon and all the other stars in the firmament. it stood over them like an angry god. then suddenly there was a flapping sound, as though a great many bats had come out of the

### F.3   PYTHIA 410M   TOP-P: 0.95   REPPENALTY: 1.2   BEAMSIZE: 8

**Wikipedia Context:**  representative of the jewish consciousness.  rosenfeld noted that while holocaust literature often either soars to " expostulation and apostrophe ", or sinks to " a
**Generation:** paltry, self-congratulatory tone ", the same can be said of much of the literature of the post-holocaust period. rosenfeld also noted that the holocaust literature of the post-holocaust period was characterized by a great deal of self-congratulation, as well as a great deal of self-deprecation. for example, rosenfeld noted that the holocaust literature of the post-holocaust period was characterized by a great

---

**Wikipedia Context:** mastodonsaurus and other similar animals were referred to as labyrinthodonts, named like labyrinthodon for teeth that were highly folded in cross section
**Generation:**  .   references category:late cretaceous dinosaurs of north america category:labyrinthodonts category:fossils of the united states category:fossils of canada category:fossils of mexico category:fossils of south america category:fossils of the united states category:fossils of mexico category:fossils of canada category:fossils of the united states

---

**Wikipedia Context:** on july 9, pagasa classified a system as tropical depression falcon, off the west coast of the philippines. the jma also briefly initiated advis
**Generation:** ories on the system. on july 10, pagasa issued a tropical cyclone watch for the area of the storm. on july 11, pagasa issued a tropical cyclone watch for the area of the storm. on july 12, pagasa issued a tropical cyclone watch for the area of the storm. on july 13, pagasa issued a tropical cyclone watch for the area of the storm

---

**Fiction Context:** club, and suddenly he perceived the occasion and, rolling over, began to struggle to his feet. he stood but clumsily. he supported
**Generation:** himself with one hand on the wall, and with the other he clutched at the window-sill. "who are you?" he cried. "who are you?" "it is i," said the man. "who are you?" "it is i," said the man. "who are you?" "it is i," said the man. "who are you?" "it

---

**Fiction Context:**  in the same grey covering.  then i perceived, standing strange and gaunt in the centre of the hall, what was clearly the lower part of a
**Generation:** woman's body, and i knew that it was the body of a woman. i did not know who she was, but i knew that it was the body of a woman. it was the body of a woman, and it was the body of a woman, and it was the body of a woman, and it was the body of a woman, and it was the body of a woman, and it was the body of a woman, and it was

---

**Fiction Context:** a time my brain went stagnant. presently i got up and came through the passage here, limping, because my heel was still painful, and
**Generation:** i couldn't stand it any longer. "i'm sorry," i said, "but i can't stand it any longer. i'm going to get out of here." "you can't get out of here," she said. "yes i can," i said. "no, you can't," she said. "yes i can," i said. "no, you can't," she

### F.4 PYTHIA 1B TOP-P: 0.95 REPPENALTY: 1.2 BEAMSIZE: 8

**Wikipedia Context:** hünigen, the battles at ostrach and stockach, and the northern swiss campaign of 1799 that included battles at winterthur and zürich
**Generation:** . as a result of his service in the war of the austrian succession (1740–48), he was made a knight of the order of the golden fleece. references external links category:1743 births category:1802 deaths category:austrian generals category:austrian military personnel of the war of the austrian succession category:recipients of the order of the golden fleece category:

. ______________________________________________

**Wikipedia Context:** from his base in tel aviv, holocaust survivor emmanuel lieber directs a group of jewish nazi hunters in search of adolf hitler. lieber believes that the former
**Generation:** nazi leader is hiding in plain sight in the united states. lieber's film is based on the true story of the hunt for adolf hitler in the united states during world war ii. in the film, lieber travels to the u.s. holocaust memorial museum in washington, d.c. to meet with holocaust survivors and descendants of those who participated in the hunt for adolf hitler. lieber's film will be shown at

. ______________________________________________

**Wikipedia Context:** following the release of the album, stone sour went on to promote it for over a year ; releasing five singles and touring in several regions, including the united
**Generation:** states, canada, mexico, and the united kingdom. after the release of the album, stone sour went on to promote it for over a year ; releasing five singles and touring in several regions, including the united states, canada, mexico, and the united kingdom. after the release of the album, stone sour went on to promote it for over a year ; releasing five singles and touring in several regions, including the united states, canada, mexico,

. ______________________________________________

**Fiction Context:** then came the bats, and the star that was like ugh-lomi crept out of its blue hiding-place in the west. she called to
**Generation:** him, but he did not answer, and she went on calling, but he did not answer, and she went on calling, but he did not answer, and she went on calling, but he did not answer, and she went on calling, but he did not answer, and she went on calling, but he did not answer, and she went on calling, but he did not answer

. ______________________________________________

**Fiction Context:** looks more like the first half of an animal than anything else in the world," he said. "fore-legs and no hind." "
**Generation:** what do you mean by fore-legs?" i asked. "the fore-leg is the part of the body that is closest to the ground, and the hind-leg is the part of the body that is farthest from the ground." "what do you call the hind-leg?" i asked. "the hind-leg is the part of the body that is farthest from the ground, and the fore-leg

. ______________________________________________

**Fiction Context:** some months, and he was thinking of "reducing the figure," when he made a singular discovery. at that time his health was very bad
**Generation:** , and he was obliged to give up his profession, which he had been practising for more than twenty years, and to return to his old home, where he had lived for the last ten years. he was then about fifty-five years of age, and he was a man of great strength of body, but his mind was feeble, and he was subject to many diseases. he had been in the army, and had served

## F.5  LLAMA2 7B  TOP-P: 0.95  REPPENALTY: 1.2  BEAMSIZE: 8

**Wikipedia Context:** the irish republican army ( ira ) had been inactive militarily since the end of the border campaign in 1962. it was low in both personnel and equipment — chief
**Generation:** was not until 1969 that the provisional ira (pira) began a campaign of bombings and shootings against the british army, the royal ulster constabulary (ruc) and protestant civilians in northern ireland. in 1970, the pira bombed the old bailey in london, killing one person and injuring 200. in 1972, the pi

---

**Wikipedia Context:** the player, as donald, ventures through a variety of side-scrolling levels. generally, each level is divided into an overland part and a dungeon
**Generation:** part. in the overland part, the player controls donald from a top-down perspective, while in the dungeon part, the player controls donald from a side-scrolling perspective. in the overland part, the player's goal is to reach the end of the level, while in the dungeon part, the player's goal is to reach the end of the dungeon while collecting items and defeating enemies. ## reception reception agg

---

**Wikipedia Context:** during the withdraw, however, a company ran into a sizable north korean force and had scattered in the ensuing fight. most of the company, including its commander
**Generation:** 's officers were killed or wounded, and the survivors found themselves cut off from the rest of the battalion. on the morning of june 25, 1950, the 24th infantry division's 2nd battalion, 34th infantry regiment, commanded by lt. col. charles b. smith, was ordered to relieve the beleaguered a company of the 19th infantry regiment.

---

**Fiction Context:** somehow, to feel any humanity in the things. only my disinclination to leave weena, and a persuasion that if i
**Generation:** did i should never find her again, prevented me from doing so. it was about two o'clock in the afternoon when i returned to the spot where i had left weena. she was not there. i wandered about for some time, but could find no trace of her. at last i came to the conclusion that she must have wandered away in search of food, and that i had better return to the hut. when i reached the

---

**Fiction Context:** sunlit city of delight he knew of in japan, half-way about the world. she loved the dream, but she feared the leap;
**Generation:** she loved the man, but she feared the leap; she loved the leap, but she feared the man. she loved the dream, but she feared the leap; she loved the man, but she feared the leap; she loved the leap, but she feared the man. she loved the dream, but she feared the leap; she loved the man, but she feared the leap; she loved the leap, but she feared the

---

**Fiction Context:** began to put my interpretation upon the things i had seen, and as it shaped itself to me that evening, my interpretation was something in this way.
**Generation:** it was the middle of the afternoon, and i was sitting on the edge of the porch, with my feet dangling in the grass, and my elbows on my knees, and my chin in the hollow of my hands, and my eyes on the ground, watching the ants as they came out of the wood-pile and trailed off into the sweet-smelling grass, watching the squirrels chase each other up and down the tr

## F.6 GPT-4

**Wikipedia Context:** spacey was mendes'first choice for the role of lester, though dreamworks had urged the director to consider better-known actors ; similarly, the
**Generation:** studio suggested a number of high-profile actresses for the part of carolyn, lester's dissatisfied wife. however, the director strongly believed in his casting choices and fought for spacey and annette bening, whom he felt embodied the characters perfectly. mendes felt that spacey had the ability to portray lester's complex personality, showcasing the diametrically opposed facets of his character - the comical and the tragically flawed. spacey's effortless versatility as an actor de-cl

---

**Wikipedia Context:** osbert was no longer archdeacon by 1158, as his successor is attested by that point. osbert, however, continued to call himself
**Generation:** archdeacon" through the years. this wasn't due to delusions or an inability to accept his reality, but rather as a means of maintaining a significant influence within the clergy and his surrounding community. his knowledge, experience, and wisdom were still highly regarded and sought after, and his self-proclaimed title seemed to carry a sense of endearment rather than mockery. despite losing his official title, osbert remained an integral part of the church; he assisted in monumental projects

---

**Wikipedia Context:** new york state route 31b ( ny 31b ) was a state highway in central new york in the united states. it served as a connector between ny 31
**Generation:** and ny 96 in the town of clifton springs. the highway was designated in 1949 and ran for approximately 6.6 miles through the rural landscapes of ontario county. the term "31b" was not indicative of a sequential order in the new york state route system, but instead detailed the nature of the route as a branch of ny 31. the highway started at an intersection with ny 31 in the hamlet of gypsum and moved east into cl

---

**Fiction Context:** but his chief thought was of the crystal. he approached that topic in a gingerly manner, because he knew mrs. cave's peculiarities. he
**Generation:** had dealt with her in the past and remembered how fiercely she guarded the family heirlooms, how her eyes would sparkle with a fierce territoriality whenever anyone expressed the slightest interest in them. mrs. cave, a solid woman of uncertain age with a hairline receding much faster than her fiery spirit, could become a lioness if she perceived a threat to her treasure. but this time, it was different. this time, he needed the crystal, not just as a curiosity

---

**Fiction Context:** why, too, if the eloi were masters, could they not restore the machine to me? and why were they so terribly afraid of the
**Generation:** dark? these questions tormented me as i sat among them, overwhelmed by their alienness. the eloi, in all their gentility and grace, held a deep-rooted terror of the night. i observed a sense of dread creeping into their eyes as the sun began its descent, their laughter and merriment getting replaced by silence. not a single one of them dared to move without a flicker of terror on their beautiful, fragile faces. i couldn't understand

---

**Fiction Context:** been speculating upon the destinies of our race, until i have hatched this fiction. treat my assertion of its truth as a mere
**Generation:** figment of my overactive imagination, an effusion of my persistent curiosity about the trajectory of humanity. yet beneath all my declarations lies an abysmal concern. in the grand purview of human history, could it not be possible that my fantasy may morph into reality? could it not be that we, as an entity constantly evolving and pushing the boundaries of our knowledge, are unknowingly sowing the seeds of these imagined futures with every choice we make? my conjured world

