# OpenReview forum: "Branch-GAN: Improving Text Generation with (not so) Large Language Models"
_ICLR.cc/2024/Conference — ICLR 2024 poster_

### Official Review · Reviewer_rZ5m · 2023-10-29

**Soundness:** 3 good
**Presentation:** 3 good
**Contribution:** 3 good
**Rating:** 6
**Confidence:** 4

**Summary:**

The paper proposed an adversarial training strategy to enhance the generation ability of LLMs, it generated multiple external branches along the input sequence and add RL loss over these branches. The manual evaluation on 10k+ annotations is conducted to demonstrate the effectiveness of the proposed method.

**Strengths:**

1.A GAN-style training schema is proposed based on benefit of parallel computation of Transformer, which generates multiple branches and results in dense learning signals, this improves the model generation ability without any new training data.
2.Extensive human evaluation and analysis to demonstrate the effectiveness the method.

**Weaknesses:**

1.Both generator and discriminator need to be tuned. This results in expensive computation and memory consumption, especially for large LLMs.

**Questions:**

1.For different generation tasks (such as Wiki, Fiction generation), was the proposed method tuned separately? In other words, did the current version support multi-task tuning?
2.For sequence S with length N, K unique starting points is randomly sampled from [1, N-1] as branches. As each generated branch consists of d (e.g., 16) tokens, if the sampled starting point is N-1, is the corresponding generated branch easy to be recognized than that corresponding to other points? Or should we sample the starting point from the range [1, N - d]?
3.For each branch, if the termination token was generated, was the branch still generated until its length = d?
4.How about the training complexity w.r.t. branch number and depth of generation?
5.For robustness analysis, how to repeat the random selected characters? It is better to give an example. Is the noise sensitive to the position of selected characters?

---

> ### Author Response · Authors · 2023-11-16
> **Response to Reviewer rZ5m**
>
> Thank you for your time, review and curiosity!
> Clarifying these questions will surely improve the quality of the paper.
>
> **Regarding the main weakness of having to tune both a generator and discriminator.**
> It is true that the current bare-bones implementation of Branch-GAN always tunes both models. However, we do not agree that this diminishes the academic contribution, and is thus not a weakness of the paper in itself. Rather we find that introducing these concepts in a straight-forward manner to be a strength.
>
> That being said, the increased per-sample computational cost pales in comparison to that of the original pre-training. Consider that we utilize less than 0.01% of the original training data. Due to the efficient encoding method, training with a depth of 16 means we perform 16 extra forward passes per sample. We train two models (Gen + Disc) for 4 epochs. giving us: 0.01% x 16 x 2 x 4 = 1.28%
> In a very over-simplified manner, this means that all our training is roughly 1.28% of a **single** epoch of the pre-training.
>
> For our biggest model this training can be comfortably executed on a single DGX server (8 GPUs) under 48 hours. We fully agree that this deserves further explanation in the paper, and we have created the additional Appendix section D.1.
>
> **Regarding Question-1:**
> No, the Branch-GAN model was not tuned separately for the different tasks. There is just the continued training on the 0.01% of the Pile dataset. Paragraph 2 in Section 4.1 has been updated to more explicitly explain this.
>
> **Regarding Question-2:**
> A very astute question! Indeed the discriminator could easily learn to distinguish tokens that surpass N. In our experiments any token that exceeds N is automatically masked, and yields no loss for either the generator and discriminator. This is mentioned in the second paragraph of Section 3.3
>
>
> Only sampling starting points in the range [1, N-d], as you propose, would ensure that compute is not wasted on branches exceeding N. If further work can demonstrate that it does not introduce any detrimental bias in the training it would indeed be preferable. We avoided this since it means that the last d tokens would have less probability of being generated than with our masking schema.
>
> **Regarding Question-3:**
> Admittedly, we have not considered this possibility. It would make sense that the loss is masked after the termination token. Something to be incorporated in future work, thank you!
>
> **Regarding Question-4:**
> We are not sure exactly what you’re asking for, as training complexity is somewhat nebulous. Could you elaborate upon your question?
>
> The depth and number of branches explored in Section 4.6 affects training complexity in different ways. The number of branches mainly affects the memory load for each sample. The depth increases both the memory, but also the sequential time it takes to compute a single sample.
>
> **Regarding Question-5:**
> This is a fair request. We will create an example of the noise generation process shortly and notify you when we have upload an updated version.

---

> > ### Author Response · Authors · 2023-11-20
> > **Response #2 to Reviewer rZ5m**
> >
> > We have now added Appendix E.4 where we explain the noise generation in more detail and show some examples

---

> > > ### Comment · Reviewer_rZ5m · 2023-11-22
> > >
> > > Thanks for the authors' detailed response and also giving explanation to most of the questions. The added Appendixes are helpful to understand the proposed method well. I would like to hold my evaluation scores.

---

### Official Review · Reviewer_MWH7 · 2023-11-01

**Soundness:** 3 good
**Presentation:** 4 excellent
**Contribution:** 3 good
**Rating:** 6
**Confidence:** 5

**Summary:**

The paper presents Branch-GAN, a novel method that improves text generation quality in large language models (LLMs) by using adversarial training. It overcomes the deficiencies of current models such as repetitive texts and looping issues by generating multiple branching sequences from each training sample, providing dense signals for both generator and discriminator. This leads to stable training dynamics and high-quality text generation, using only a fraction of the training data typically required. Human evaluations confirmed that Branch-GAN outperforms much larger baseline models and has robustness against noisy inputs. The paper concludes that adversarial training enhances LLMs' capabilities and suggests further exploration into adversarial learning for increased data efficiency and application to larger models.

**Strengths:**

Advantages:

 - Higher Quality Text Generation: Branch-GAN can generate texts of significantly better quality on average compared to other models, as confirmed by human evaluations.
 - Data-Efficient Fine-Pretraining: The method achieves this enhanced performance while utilizing less than 0.01% of the original training data, indicating exceptional data efficiency.
 - Stable Training Dynamics: By generating multiple branching sequences from each training sample and processing them in parallel, Branch-GAN provides a dense and stable learning signal, leading to more stable training dynamics compared to existing language GANs that leverage sequence-level rewards (e.g. SeqGAN, RankGAN etc). I especially like the idea of using three prediction heads such that it implicitly incorporates the benefits of scheduled sampling, which is an algorithm that is known to practically help the model produce better outputs, despite some criticism from How (not) to train your generative model [https://arxiv.org/abs/1511.05101 ].

**Weaknesses:**

Disadvantages:

 - Confusing Illustrations: While many of the important elements have been shown in Fig1, I would really appreciate if the authors can considering showing some technical details of the discriminator in Fig 1 too. With the current illustration it's really difficult to know enough details about how the dense adversarial rewards are constructed, which are actually the most novel part of the proposed method. (The efficient pretext encoding reusing trick is good, but it's not what I think that differentiates BranchGAN from other language GANs)

 - Lacking Mode Collapse Detection and/or Diversity Measure [Addressed during Rebuttal]: According to some previous criticism against language GANs from Language GANs Falling Short [https://arxiv.org/abs/1811.02549 ], most previous language GANs that have shown effectiveness in producing more coherent output sequences generally can be fragile against the comparison of simply annealing the temperature of language models. In other words, it could be the case that BranchGAN, too, sacrifices diversity/mode coverage in exchange for precision/quality. A proper study on this is needed to respond to such challenges. **This is dangerously needing attention** because the remarkably increased PPL of the BranchGAN-generated samples by Pythia 6.9b and remarkably increased generation quality reminds me of some previous language GANs that also over-claim themselves to be able to generate high-quality samples, eventually turned out to be rather trading-off between quality and diversity and had zero improvement when considering both aspects.

**Questions:**

See Weaknesses)

**Details Of Ethics Concerns:**

The dataset the models (Pythia/GPTNeoX and the BranchGANs finetuned from them) use, known as the Pile dataset, is now under controversy for its violation of copyright. It has been taken down (up to Oct. 2023) from the open Internet. While this does not affect the academic values of the paper, the authors and ICLR should be careful when open-sourcing the trained model parameters.

---

> ### Author Response · Authors · 2023-11-19
> **Response to Reviewer MWH7**
>
> Thank you for your review! It brings some important contributions.
>
> The necessity of additional experiments is well received, and we have updated the paper to accommodate your concern. Considering your expertise regarding Language GANS, and your appreciation of our other experimental results, we hope these new results convince you of the validity and publishability of our contributions.
>
> **Regarding Lack of Diversity Measure:**
> We are grateful for this important observation. Evaluation results in this setting seems indeed crucial given the prior work and criticism. To rectify this oversight Appendix E.4 now contains a Diversity-Quality experiment with a temperature sweep for selected models according to the proposed experiment.
>
> That being said, we struggle with the intuition of the evaluation process of the proposed paper. Measuring quality via the BLEU overlap between generated texts and a given test set seems infeasible in open-ended text generation. Indeed, as one can tell from results in Appendix E.4 there is a clear contradiction in the perceived quality given the BLEU metrics and our human evaluation.
>
> Faced with these contradicting results, we carefully tried to find any bugs in our code, and instead decided to using the accompanying code of the “Language GANs Falling Short” paper: https://github.com/pclucas14/GansFallingShort/blob/master/real_data_experiments/eval_bleu.py
>
> However, the contradicting results still remained, and upon further investigation we found others who questioned this evaluation process or the metrics being used. In
> https://arxiv.org/pdf/1806.04936.pdf they point out some limitations of automatic evaluation scores for diversity and quality. Additionally, https://arxiv.org/pdf/1905.09922.pdf mentions a model that generates very poor quality samples yet still received close to a perfect BLEU-5 score.
>
> We acknowledge the merit of evaluating text generation models over a sweep of temperatures. Especially if prior work has found this to be a weak point of GAN’s. But, we recognize that the automatic evaluation in these experiments are somewhat controversial. Thus, we hope that you find our results, which include human evaluation, to be a worthy contribution to this ongoing discourse.
>
> **Regarding Confusing Illustrations:**
> Your intuition about the benefits of the dense reward is fully aligned with ours. We did admittedly struggle to properly illustrate all we wanted within the page limit. Considering that both the dense reward and the pretext coding needs to be introduced, would you be satisfied with a more extensive illustration in the Appendix that covers both the generator and discriminator? If so, then we will do our best to have one done prior to the rebuttal deadline.
>
> **Regarding Ethical Concerns:**
> The current discourse regarding the Pile dataset is news to us! Thank you for highlighting this. Luckily training with Branch-GAN is comparably quick. If the discourse is not resolved before publication we will train new models on other data.

---

> > ### Comment · Reviewer_MWH7 · 2023-11-20
> > **Further response**
> >
> > Thank you my dear authors! I appreciate all your efforts in further investigation. Actually there's a simple diversity measure that you can consider - simply reporting the NLL of the BranchGAN model by using it as an LM on the dev/test split of your data would be sufficient. As long as we can show that, with reasonable or zero (I dont expect to see the latter though - it is generally too difficult to be true for language GANs) **probabilistic recall** of the dev set samples, BranchGAN can significantly improve the generation quality *i.e.* the **probabilistic precision** (which I am already convinced by the currently reported results), I would buy BranchGAN as a very impactful work that can renovate people's passion in *making language GANs great again*.
> >
> > I believe these results are rather easy to be collected and reported. I'd be more than happy to raise my rating to 6 or 7 if these results could be reported in the next revision.
> >
> > If you should have any other questions or would like me to give you some further details, please let me know. I generally really like this work and am really eager to help make it a solid work that lasts for years to come.

---

> > > ### Author Response · Authors · 2023-11-20
> > > **Second Response to Reviewer MWH7**
> > >
> > > Your acknowledgement of our rebuttal efforts is well received!
> > >
> > > **Regarding NLL over test sets:**
> > > Given your advice we have now created **Appendix E.5**, which contains the NLL for various different datasets. We hope this provides the information you want. The results, mirroring the PPL results in Section 4.4, are indeed very interesting. As elaborated in the new Appendix section, we find Branch-GAN to achieve a worse absolute NLL score but a preferred relativistic NLL score when factoring in texts that were rated as *“bad”* by humans.
> > > Considering the strong human preference of texts generated by Branch-GAN, we think this indicates an interesting avenue for future research.
> > >
> > > *Let’s make Language GANs great again!*

---

> > > > ### Comment · Reviewer_MWH7 · 2023-11-20
> > > >
> > > > Could you please check the revision? I don't see the new E.5. It looks identical to the last version after the first rebuttal revision.

---

> > > > > ### Author Response · Authors · 2023-11-20
> > > > > **Wops**
> > > > >
> > > > > Ahh, our misstake!
> > > > > Please check again now.

---

> > > > > > ### Comment · Reviewer_MWH7 · 2023-11-21
> > > > > >
> > > > > > I saw these results. Well I must say I am still a little bit disappointed, but I appreciate the efforts.
> > > > > >
> > > > > > As is promised, since the *probabilistic recall* is not compromised by an unacceptable margin, I'm ready to raise my score from 5 to 6.

---

### Official Review · Reviewer_Hqg8 · 2023-11-01

**Soundness:** 2 fair
**Presentation:** 3 good
**Contribution:** 3 good
**Rating:** 6
**Confidence:** 3

**Summary:**

This paper presents Branch-GAN, a method for improving text generation with Transformers using generative adversarial networks (GANs). Branch-GAN generates multiple branching sequences from each training sample, and trains the generator and the discriminator in parallel. The paper shows that Branch-GAN models produce higher quality texts than large language models (LLMs) and other baselines, according to human evaluation and automatic metrics. The paper also explores the effects of noise, depth, and sparsity on text generation, and discusses the potential applications and limitations of Branch-GAN.

**Strengths:**

1. **Novel method**: A novel method for training Transformers in a GAN setup, which improves text generation quality with fewer model parameters and less data. I really like this idea to branch the sequence for efficient discrimination.
2. **Human evaluation**: A large-scale human evaluation study that shows that Branch-GAN models generate significantly better texts than LLMs and other baselines.
3. **Robustness**: An analysis of the robustness of Branch-GAN models to noisy contexts, showing that they can generate texts with stable TTR and perplexity.

**Weaknesses:**

1. **Lack of comparison**: The paper does not compare Branch-GAN with other language GANs or cooperative language GANs that use pre-trained Transformers.
2. **Limited evaluation**: My main concern is that the paper only evaluates text generation on two domains (Wikipedia and Fictional Stories) and does not test the generalization ability of Branch-GAN on other domains or tasks, such as summarization, dialogue, or question answering.

**Questions:**

How well does Branch-GAN generalize to other domains or tasks, such as summarization, dialogue, or question answering? What are the challenges or limitations of applying Branch-GAN to these scenarios?

---

> ### Author Response · Authors · 2023-11-16
> **Response to Reviewer Hqg8**
>
> Thank you for your time and review.
>
> We hear and recognize your concerns, to which we commit new experimental data in combination with arguments. Hopefully, after reviewing these changes and insights you will consider rating our paper higher.
>
> **Regarding your main concern of limited evaluation:**
> We fully share your curiosity of further evaluation on other tasks. The current evaluation is mainly due to space limitations, and that previous conference papers have seemingly focused on a single task when introducing new algorithmic concepts.
> Therefore, we decided on the very general task of open-ended text-generation, and to evaluate it as rigorously as possible. Hoping that this would give us enough space to properly introduce our ideas.
>
> However, we can at least partially accommodate your concern of only performing text generation on two datasets. Appendix E.3 now contains a table that evaluates the relevant models on 5 additional datasets using the two automatic metrics we established in the paper. This means that in total we now evaluate on the following types of datasets:
>
> Wikipedia, Stories, CC News, Dialogsum, Legal Summarization, Medical transcripts, OpenWeb
>
> **Regarding lack of GAN comparisons:**
> To the best of our knowledge, no previous GAN has shown improvements on long open-ended text generation, compared to that of SOTA LLMs’. Indeed, as our human study shows, even 7B LLMs struggle to generate convincing sequences of 96 tokens.
> Considering that no GAN is seemingly used for these scenarios, we opted on only evaluating LLMs (not to mention how relatively compute expensive it would be to apply other GANs to this sequence length.)
>
> **Regarding the question of generalisation to other tasks:**
> There is no inherent limitation to applying Branch-GAN to any particular domain or task. As now depicted in Appendix E.3, the same Branch-GAN models seemingly perform fairly well across a wide range of datasets.
>
> That being said, for tasks such as summarization or translation, it would admittedly be beneficial to only generate branch sequences from the summarized/translated text tokens. This means that one would change the algorithm to apply loss and generations only to the target translation tokens.

---

> > ### Comment · Reviewer_Hqg8 · 2023-11-22
> >
> > I've read the author response and decided to keep my original recommendation.

---

### Official Review · Reviewer_RVL4 · 2023-11-02

**Soundness:** 3 good
**Presentation:** 3 good
**Contribution:** 3 good
**Rating:** 8
**Confidence:** 3

**Summary:**

This paper introduces a new strategy for training language models using a combination of adversarial and MLE style losses. Historically, Language GANs have been hard to get to work well---successful methods have mainly resorted to primarily relying on MLE losses with a minor amount of adversarial fine-tuning. This work tries to alleviate some of the optimization issues for discrete language GANs by using a efficient training strategy that simultaneously optimizes multiple generations that branch off of a supervised sequence at various points in time. Combined with a MLE + dense rewards criterion, the paper shows that an effective language models can be learned.

**Strengths:**

The paper is clearly written, and the approach is simple while also appearing to be very successful. The results are also well-evaluated (at least for the specific text-continuation setting that is focused on). More broadly, I think that the paper introduces an interesting efficient batching technique for training language models with reinforcement learning --- that can possibly be applicable to other types of rewards beyond GANs (e.g., such as RLHF models).

The dataset of manual text quality rankings also seems like a valuable contribution if released, and the analysis on correlation (or lack thereof) of automatic metrics with human judgements could be of interest to practitioners.

**Weaknesses:**

Though the human evaluation on text completion is good, I'm really quite curious to see how the Branch-GAN performs on tasks that are more divergent from the pre-training setup, such as prompting tasks.

I also find the loss function in Eq. (1) to be poorly motivated. Why choose this variant, over say REINFORCE/PPO with a baseline/advantage function? (I would also at least move more of the description of the loss in A.1 to the main text.)

**Questions:**

- I'm not clear about Table 4 in the appendix --- which model is being evaluated? I would have thought Branch-GAN, but don't understand why it would be compared "vs. Branch-GAN 1B*".
- I'm also curious about win-rate _between_ models (it's noted in Sec. 5 that GPT-4 is consistently preferred over Branch-GAN, but I could only find comparisons to original texts in the paper).

---

> ### Author Response · Authors · 2023-11-16
> **Response to Reviewer RVL4**
>
> Your review and encouragement is well received, thank you.
>
> We share your enthusiasm about applying this attention mechanism and Branch-GAN to other areas such as RLHF. Indeed, all the human evaluation data will be made publicly available, and we believe it to be useful for various research contexts.
>
> **Regarding evaluation on text completion only.**
> We fully share your curiosity of evaluation on other tasks. Considering the many new concepts that had to be condensed into the main paper, it seemed natural to focus on a single task evaluation. Future work should definitely explore other areas such as summarization, translation, instruction tuning and so on.
>
> **Regarding the motivation of the loss function.**
> Due to spatial reasons we are currently struggling to squeeze more motivation regarding the loss function into the main paper. PPO and REINFORCE were evaluated in early stages of development, but lead to a very unstable training dynamic. The motivation for having two value heads is mainly intuitive, as explained in Appendix A.1, but since switching to this loss we encountered zero problems with collapse of either the generator or discriminator. Appendix A.2 has been created to share these insights in the paper.
>
> **Regarding table-4 in appendix.**
> Table 4 contains the aggregated data from all collected annotations. We have updated the title and description to clarify this further.
>
> **Regarding win rate between models.**
> This table was not included in the main paper due to spatial reasons, but we see no reason to not include it. Clearly an oversight on our part, thank you. It now exists in appendix E.1

---

> > ### Comment · Reviewer_RVL4 · 2023-11-22
> >
> > Thank you for responding to my comments, and the added results. I will keep my score.

---

### Meta-Review · Area_Chair_oa2y · 2023-12-14

**Metareview:**

This paper proposes an adversarial fine-tuning strategy for pretrained LLMs. The model generates multiple branching continuations of training examples, and then updates model parameters based on the combination of an adversarial loss and the traditional autoregressive cross-entropy loss. The resulting approach leads to improved language generation results in human evaluations. Reviewers are all in favor of acceptance, citing strengths that include the model's high performance in human evaluations and the novel branch-based adversarial training setup. The main weakness brought up were (a) the lack of evaluations on a more diverse language tasks and (b) concerns about mode collapse during adversarial training. These concerns were mostly addressed in rebuttal. The authors conducted new experiments showing an increase in perplexity on held-out data after adversarial training (a bad thing), but not a devastating increase -- mitigating concerns about mode collapse to some extent.

**Justification For Why Not Higher Score:**

While the method proposed is new and resolves some issues with adversarial training of LMs, it does seem to substantially increase LM perplexity on held-out data.

**Justification For Why Not Lower Score:**

Through its novel branching setup, the method solves some existing problems with adversarial training of LMs and will be of general interest given recent efforts to discover new ways of training and fine-tuning LLMs.

---

### Decision · Program_Chairs · 2024-01-16

Accept (poster)